# Evaluation of Optimized Preprocessing and Modeling Algorithms for Prediction of Soil Properties Using VIS-NIR Spectroscopy

**DOI:** 10.3390/s21206745

**Published:** 2021-10-11

**Authors:** Rebecca-Jo Vestergaard, Hiteshkumar Bhogilal Vasava, Doug Aspinall, Songchao Chen, Adam Gillespie, Viacheslav Adamchuk, Asim Biswas

**Affiliations:** 1School of Environmental Sciences, University of Guelph, Guelph, ON N1 G2W1, Canada; rvesterg@uoguelph.ca (R.-J.V.); hvasava@uoguelph.ca (H.B.V.); agilles@uoguelph.ca (A.G.); 2Woodrill Farms Ltd., Guelph, ON N1H 6H8, Canada; daspinall@woodrill.com; 3ZJU-Hangzhou Global Scientific and Technological Innovation Center, Hangzhou 311200, China; chensongchao@zju.edu.cn; 4Department of Bioresource Engineering, McGill University, Ste-Anne-de-Bellevue, QC H9X 3V9, Canada; viacheslav.adamchuk@mcgill.ca

**Keywords:** proximal soil sensing, precision agriculture, digital soil maps, soil characterization, soil core profiles

## Abstract

The absorbance spectra for air-dried and ground soil samples from Ontario, Canada were collected in the visible and near-infrared (VIS-NIR) region from 343 to 2200 nm. The study examined thirteen combination of six preprocessing (1st derivative, 2nd derivative, Savitzky-Golay, Gap, SNV and Detrend) method included in ‘prospectr’ R package along with four modeling approaches: partial least square regression (PLSR), cubist, random forest (RF), and extreme learning machine (ELM) for prediction of the soil organic matter (SOM). The 1st derivative + gap, 2nd derivative + gap and standard normal variance (SNV) were the best preprocessing algorithms. Thus, only these three preprocessing algorithms along with four modeling approaches were used for prediction of soil pH, electrical conductively (EC), %sand, %silt, %clay, %very coarse sand (VCS), %coarse sand (CS), %medium sand (ms) and %fine sand (fs). The results showed that OM, pH, %sand, %silt and %CS were all predicted with confidence (R^2^ > 0.60) and the combination of 1st derivative + gap and RF gained the best performance. A detailed comparison of the preprocessing and modeling algorithms for various soil properties in this study demonstrate that for better prediction of soil properties using VIS-NIR spectroscopy requires different preprocessing and modeling algorithms. However, in general RF and 1st derivative + gap can be labeled at the best combination of preprocessing and modelling algorithms.

## 1. Introduction

The global food demand of an increasing population poses tremendous pressure on our limited land resources. This calls for an improved and efficient management of soil, one of the three most important natural resources, which requires detailed information (FAO [1]). Increasing demand for soil data in agriculture has brought the need for a timely and cost-efficient method of soil analysis [2]. Soil data is used by farmers to make informed decisions on what crops they grow and what inputs they use [3]. Traditionally, several soil samples across the sampling area or a field are collected and sent to the laboratory for analysis which can be a lengthy and costly process. Owning to the inherent nature of the soil variability, a large number of samples following an intensive sampling strategy are required to characterize the variability in an agricultural field [4].

Spectroscopy, sensing the reflectance of electromagnetic radiation (EMR) from the soil’s surface [5] offers a promising alternate approach for rapid prediction of soil properties [2]. Soil properties influence the reflectance of light at diagnostic wavelengths, soil spectroscopy can be used to simultaneously estimate several soil properties [5].The soil organic carbon (SOC), texture, pH, and EC were among the most commonly predicted soil properties in the literature (Table 1). An average prediction accuracy (R^2^) using VIS-NIR for SOC, sand, silt, clay, pH and EC were reported 0.79, 0.70, 0.59, 0.76, 0.61 and 0.38 respectively [6]. 

Several different algorithms were used in these studies for preprocessing and modelling of spectral data. Zhang et al. [2], Gholizade et al. [11] and Leone et al. [12] used the Savitzky-Golay algorithms for the preprocessing of spectral data. Zhang et al. [2] and Leone et al. [12] also used standard normal variance (SNV) algorithm for preprocessing. While Terra et al. [10] did not preprocess the spectral data before modelling. Viscarra Rossel et al. [14] did not include comparisons of different preprocessing algorithms, however, they did compare modelling algorithms, including partial least squares regression (PLSR), principle component analysis, stepwise multi-linear regression (SMLR) and nearest neighbor modelling algorithms which can all be found in the review Viscarra Rossel et al. [14]. Additional modelling algorithms found in the literature include: (i) random forest (RF): This model tries to take benefit from random feature selection in addition to bagging. When growing a tree in a random forest, each node is split utilizing a best selection amongst a subset of features picked randomly at that node. Decision trees are grown until a specific number of nodes is reached which can be predetermined by the user [16]; (ii) cubist: It is a prediction-oriented rule–based regression model which is a combination of ideas of Quinlan’s M5 model tree wherein the prediction depends on terminating leaves consisting of linear regression models [17], and (iii) extreme learning machine (ELM) is preferred approach in batch learning, sequential learning and incremental learning because of its rapidness and generalization ability. The approach is popular in recent time in the spectroscopic modeling for classification, regression and estimating SOM [18,19]. Variation in R^2^ values could be attributed to the range of preprocessing algorithms and modelling algorithms used. 

Preprocessing algorithms are used to normalize spectra, enhance relevant spectral fingerprint regions, and remove any physical noise before modelling of the spectral data [12,20]. Several different algorithms have been used for preprocessing of spectral data. Commonly used preprocessing algorithms include moving averages, binning, smoothing such as Savitzky-Golay filtering, normalization, continuum removal, derivatives, gap derivatives, multiplicative scatter and SNV computation [20,21,22,23]. Savitzky-Golay is a smoothing function which reduces noise by using a weighted sum of neighboring values, while derivatives remove additive or multiplicative effects between spectra [20]. The SNV normalizes data to reduce light scatter effects [20]. Preprocessing models can be used alone to focus on a specific correction, or they can be used in combination to correct more than one area of the data. For example, adding a gap to a derivative can help to smooth any noise created from the derivative itself [20]. Individual data sets, when processed with different preprocessing and modelling algorithms can have varying results; thus, it is important to determine the combination best suited to the data set. Although the literature demonstrates the use of several preprocessing algorithms, to the best of our knowledge no study used multiple preprocessing and their combination on a single data set. This study would assist researchers in selecting the optimal preprocessing algorithms to use when using spectral data for predicting soil properties. 

Modelling is an important part in the success of the spectroscopic predictions. Generally, spectral data is used against some known values of soil properties form laboratory analysis to develop a predictive relationship using various multi-variate statistical analysis. The model can then be used to predict the attribute using spectral data acquired from a soil sample. Two types of models are used in spectral predictions; statistical-based models and machine learning-based or algorithmic models [24]. Research comparing combination of preprocessing and modeling algorithms on single spectral dataset for prediction of various soil properties are very rare and mostly studying SOC or soil clay content [21,23,25,26]. Statistical-based models are based on assumptions made by the user; while machine learning models are data driven and learn from the data set without the user assuming any parameters [27]. Statistical models may limit the user’s ability to deal with statistical problems in the data, where with machine learning models the data and any problems or trends associated with it will guide the solution [27]. Some commonly used statistical modelling algorithms are PLSR and PCA. The RF and ELM are examples of the machine learning-based data-driven approach and are less commonly used modelling algorithms, but have shown promising results in spectroscopy [24]. We could not find a study which used combinations of preprocessing and modelling algorithms on a single spectral data set for the prediction of soil properties. A study of this nature is needed to determine which combination of preprocessing and modelling algorithms are optimal for the use in analyzing spectral data. 

Limited research has also been completed on the use of VIS-NIR spectroscopy in Canadian soils. The large area of glacial deposits in Canada has greatly impacted the development of its farmland [28]. Highly variable soils have developed in Canada and Ontario due to the occurrence of multiple glaciations. The diversity of the soils available in Ontario make it suitable for testing VIS-NIR spectroscopy, as we will be able to determine how VIS-NIR spectroscopy predicts on a variety of soils. The goal of this research was: (1) to examine the suitability of VIS-NIR spectroscopy to predict soil properties up to 1 m in depth using laboratory processed and airdried samples; (2) optimize various preprocessing and modeling algorithms and evaluate their performance in predicting soil properties.

## 2. Materials and Methods

### 2.1. Study Area and Sample Collection

This study was conducted on 13 cash crop farms, located in Ontario, Canada and managed by Woodrill Limited (Guelph, ON, Canada) (Figure 1). Twelve of the farms are located within Wellington County, while 1 farm is located with Dufferin County. Wellington County is comprised of a variation of soils, including 12 catenae which are made up of 39 different soil series; while Dufferin County is comprised of 21 catenae which are made up of 43 soil series [29,30].

Wellington and Dufferin Counties are both topographically diverse areas, shaped by repeated glaciations in history. Sandstone, limestone and shale bedrock can all be found underlying the soils of Wellington and Dufferin Counties [29,30]. Surface deposits of till, outwash, kame, esker, deltaic and lacustrine can also be found. Variation in physiographic features is also seen within the counties including spillways, eskers (gravel ridge), kames (sandy hill), drumlins and swamps [29,30]. An average temperature ranges in the areas from −6.6 °C to 20.0 °C on average; however, the lowest of −31.9 °C and highest of 36.5 °C have been recorded [31].The average yearly rainfall in the area is 916.5 mm and the average yearly humidity is 87.8% [31].

A total of 205 sample points within the 13 farms were pre-selected by the soils team at Woodrill Ltd. Predictive digital soil mapping procedures were used to segment each farm into soil management zones using a unique combination of topographic, crop performance and apparent electrical conductivity parameters. Raster cells with the highest membership values for a soil management zone were selected for core sampling (Doug Aspinall and Dan Breckon, personal communication, 30 May 2019).

Soil profiles were collected with a Post Pounder (Deer Fence Canada, Dunrobin, ON, Canada) that was modified to drive a reinforced 120 cm steel coring tube fitted with a 4.5 cm diameter plastic insert (Figure 2a) (Doug Aspinall and Dan Breckon, personal communication, 30 May 2019). The soil sampling was carried out between August and October for both 2016 and 2017. The cores were labelled, capped, and stored in a cool dark room prior to analysis. A soil profile description was completed for each core during the winters of 2016 and 2017. The soil core was placed into a trough (half piece of PVC pipe) and the plastic insert was cut carefully on 2 sides to minimize any smearing. The top half of the plastic insert was carefully removed from the soil core. Next, the core was gently rolled onto a sliding table and then split into two to expose the soil profile. The soil profile description included horizon names, upper and lower horizon depth of the horizons, parent material, hand texture assessment and drainage class. A soil type name was assigned to the profile after the profile description was completed (Doug Aspinall and Dan Breckon, personal communication, 30 May 2019). The soil horizons were classified according to the Canadian System of Soil Classification [32]. Drainage classification and hand texture were determined using the Field Manual for describing soils in Ontario [33]. The horizon images were later stitched together (Hugin-panorama photo stitcher) to create full profile images (Figure 2b) (Doug Aspinall and Dan Breckon, personal communication, 30 May 2019). Each horizon was bagged, labelled, and taken to the laboratory for further analysis. In total 1046 horizon samples were collected.

### 2.2. Laboratory Methods

The soil pH was measured adopting method by Thomas [34] using a Fisher Scientific Accumet AE150 soil pH meter (Fisher Scientific, Hampton, NH, USA). The soil EC was measured using a Fisher Scientific Accumet XL600 (Fisher Scientific) according to methods outlined by Rhoades and Oster [35]. The SOM was estimated using loss on ignition (LOI) modified from Veres’s study [36] and OM was calculated using the equation below: (1)$$SOM\left(\%\right)=\frac{W_{i}-W_{f}}{W_{i}}\times 100$$
where *Wi =* Initial weight of soil sample and *Wf =* Final weight of soil sample. The soil texture analyzed using modified sieve methods from Gee and Bauder [37] and hydrometer. The clay and silt percent were determined hydrometer method, while sand fractions (1–2 mm very coarse sand (VCS), 0.5–1 mm coarse sand (CS), 0.25–0.5 mm medium sand (ms), 0.05–0.25 mm fine sand (fs)) were determined through sieving (Standard sieves #18, 35, and 60). The percent very fine sand (vfs) could not be calculated separately and was included in the overall sand portion. Due to some errors, we could measure pH, EC, and OM for 1041,1038 and 1025 soil samples respectively. Texture analysis was completed on a subset of 238 samples, sand fractions could only be determined on 230 or the 238 samples due to sieving errors.

### 2.3. Spectral Collection

Three spectral scans were taken on each air-dried and ground (<2 mm) sample. The spectrometer consists of two sensors: (i) USB2000 spectrometers (Ocean Optic Inc., Dunedin, FL, USA) covering the visible (VIS) spectrum 342 to 1023 nm with a resolution of 6 nm; (ii) a C9914GB Mini-Spectrometer (Hammatsu Photonics K.K., Tokyo, Japan) covering the spectral range of 1070 to 2220 nm with a resolution of 4 nm. The instrument has its own li halogen light source (2700 K) were used to collect the spectral data. Samples were tightly packed in a petri-dish and held directly against the spectrometer’s light to ensure no outside light would interfere with the reading. The three scans were taken from different areas of each air-dried and ground (<2 mm) soil sample to ensure an accurate representation of the sample, an average of these three-scan used for spectral analysis.

### 2.4. Optimization of Data Processing

The first step in processing of spectral data involved data cleaning to reduce the existing noise. All spectral data below 397 nm and above 2212 nm (i.e., the beginning and end of the scan) were removed to avoid any edge effects (Figure 3). 

The measurements at 1086 nm and 1092 nm were also removed, resulting in 371 spectral points with a resolution of 6 nm in the visible region and 4 nm in the near infrared region (resampled at 4 nm resolution from 6 nm raw measurements). The spectra signatures (absorbance) associated with each wavelength then used for further processing. We used the spectral data in absorbance format since it reduces nonlinearity and shows higher correlation with soil properties [2,14,38,39]. The quality of VNIR spectra can be affected by various factors such as particle size, variation of optical path, soil aggregation, moisture, and carbon content. A well-defined protocol for soil spectral acquisition aid to minimize these errors. The preprocessing methods reduce these interferences (Figure 3), thereby improving the accuracy of predictive algorithms. The commonly used preprocessing methods in soil spectroscopy are smoothing, mean centering, derivatives, normalization, standard normal variate, and multiplicative scatter correction. Organic matter data was used to optimize the preprocessing and model algorithms, as OM has shown to have the greatest correlation with VIS-NIR spectroscopy [2]. Details of six preprocessing algorithm provided in the Table 2. A few examples of the effects of preprocessing algorithms on soil spectra is shown in Figure 3 along with the original spectra collected and the average spectra used for modelling. Thirteen combinations of six preprocessing algorithms were tested in combination with 4 modeling algorithms (Table 3).

Partial least squares regression (PLSR) is a statistical-based algorithm and is the most commonly used model in spectral processing [40]. This method uses inference to model a linear relationship with the spectral data and the attribute [40,41] and is a suitable approach when dealing with missing values and data noise [14]. A detailed description on PLSR can be found in Viscarra Rossel et al. [14] and beyond the scope of this paper. Briefly, the predictor matrix *X*, where $X=\left[x_{1,}x_{2,},\ldots ,x_{i,}\right]$ was used as independent variables. Each $x_{i}$ represents one data layer from all the proximal soil sensors. Each soil property, *y*, was used as a dependent variable in PLSR, with both mean-centered. A few linear combinations (called component, or factors) *T*, of the original predictor matrix *X* were extracted. Then both *X* and *y* were regressed onto *T* as follows, $X=TP^{T}+E$, and y=Tq+f, where *P* were predictor loadings and *q* were soil property loadings, describing how the variables in *T* were related to *X* and *y*. *E* and ***f*** were residuals and represented noise or irrelevant variability in *X* and *y*. Estimated model parameters were then combined into the final prediction model as $\hat{y}={\hat{b}}_{i}x_{i}+b_{0}$ where *b_0_* was the intercept and ${\hat{b}}_{i}$ were the regression vectors.

Cubist, RF and ELM are machine learning model algorithms and have been used less frequently in spectral predictions but are of growing interest. Cubist is a unique algorithm as its predictions are not based on discrete values but are instead based on linear regression [17]. It is an extension of a tree-based model, M5 developed by [42]. A model tree is first created in this rule-based regression and then reduced to a series of rules based on spectral partition. Following this, a linear model is developed and applied to predict the target variables or soil properties. Cubist has advantages as it can utilize boosting (communities) and adjust its predictions using the neighbors withing the training dataset (neighbors). Detailed methodology on cubist can be found in Rossel, et al. [43] and Minasny and McBratney [17]. In this study, the committees and neighbors were determined using the RMSE (the lowest) in the calibration set. Leave-one-out cross validation was used to calculate the RMSE. The R package ‘Cubist’ was used for this study.

Random Forest (RF) uses decision trees and are trained by both a random subset of predicted variables and a different random data set; decision trees grow until they reach a predetermines number of nodes [16]. It is an ensemble machine learning approach that merges thousands of individual trees [44]. Each individual tree is built by bootstrapping on calibration data, and the random subspace method (the size of the subspace is denoted by *mtry*) is applied at each node split in the tree. The final prediction is the average of the predicted values from all the trees. RF generally has a better generalization ability, which is used for both regression and classification.

Finally, extreme learning machine (ELM) is a generalized single hidden layer feedforward network with a weight and first-layer hidden layer threshold and does not requires any tuning or parameter setting [45]. The thresholds in the first layer are generally randomly assigned and a least square method us used to directly calculate weight in the output layer It has an extremely fast learning speed as the whole process is completed in one round with no iterations and is more straightforward and simpler than other learning algorithms as it tends to not have issues such as improper learning rate and overfitting [45]. A simplified scheme of ELM model structure is presented in Figure 4. Detailed methodology of the methods can be found in Yang, et al. [46].

Briefly, for N distinct samples (*x_i_*, *y_i_*), where: $$x_{i}={\left[x_{i1,}\;x_{i2,}\ldots ,x_{in,}\right]}^{T}\in R^{n}$$
where, $x_{i}$ = soil spectra, and $t_{i}$ = where the observed values of target soil properties.

For given a hidden node number Ň, the activation function can be defined as follows:(2)$$g\left(x\right)=\overset{Ň}{\underset{j=1}{\sum }}\beta_{j}\;g_{j}\left(x_{i}\right)=\overset{Ň}{\underset{j=1}{\sum }}\beta_{j}g\left(w_{j}*x_{i}+b_{j}\right)=O_{i,}\;i=1,2\ldots .,N;j=1,2,\ldots Ň$$
where, $w_{j}\in R^{n}$ is the weight vector connecting the input nodes to the *j*^th^ hidden node and $\beta_{j}\in R$ is the threshold of the *j*^th^ hidden node and the output nodes. To approach the real results of the training data infinitely, the prediction result $O_{i}$ must be consistent with real result $t_{i}$, in which case $\overset{Ň}{\underset{i=1}{\sum }}\left|\left|\;O_{i}-t_{i}\;\right|\right|=0$. Under these conditions, Equation (1) can be expressed as follows $\overset{N}{\underset{i=1}{\sum }}\beta_{j}g\left(w_{j}*x_{i}+b_{j}\right)=t_{i,}$ which is represented by a matrix:Hβ=T
where:$$H={\left[\begin{array}{cc} g\left(w_{1}{\;*\;x}_{1}+b_{1}\right)\ldots & g\left(w_{Ň}{\;*\;x}_{1}+b_{Ň}\right)\\ \vdots & \vdots \\ g\left(w_{1}{\;*\;x}_{N}+b_{1}\right)\ldots & g\left(w_{Ň}{\;*\;x}_{N}+b_{Ň}\right) \end{array}\right]}_{N*Ň}$$
$$\beta ={\left[\begin{array}{c} \beta_{1}\\ \vdots \\ \beta_{Ň} \end{array}\right]}_{Ň*1}$$
$$T={\left[\begin{array}{c} t_{1}\\ \vdots \\ t_{N} \end{array}\right]}^{T}$$
where input weight $w_{j}\in R^{n}$ and bias $\beta_{j}\in R$ are randomly assigned, the output matrix H in the hidden layer can be calculated by ELM, after which the output weight *β* is calculated by *β*’ = H^+^T where H^+^ is the Mosse-Penrose generalized inverse of H.

The ‘R’ statistical package [47] was used to carry out the optimization analysis. Preprocessing algorithms were available using the ‘prospectr’ package [20]. While modelling algorithms were available in the ‘caret’, ‘cubist’, ‘elmNN’, ‘pls’ and ‘randomforest’ packages [27,45,48,49,50]. The initial spectral data was split into a 70% calibration set and a 30% validation set by Kennard and Stone method [51] and the 30% dataset was kept separate as external validation dataset. The calibration spectral dataset was further divided in to a 70% calibration and 30% as cross-validation or internal validation dataset. The calibrated model was separately tested for external validation dataset. The optimization was carried out by testing each of thirteen preprocessing and four modelling algorithms for the prediction of OM. In order to compare the performance of the optimization or the preprocessing and modelling algorithms, a series of indicators were calculated (Table 4).

Though these performance indicators were calculated during optimization process, we adopted adjusted R^2^ (R^2^_adj_) as the main criteria to compare the performance of the combinations. Based on the adjusted R^2^ values, the three best combinations were selected. The selected three best combinations were used for the prediction of all other soil properties. 

## 3. Results

### 3.1. Descriptive Statistics of Selected Soil Properties

The soil properties varied greatly within the studied fields (Table 5). For example, the range of SOM for this data set was 0.39% to 17.13%, pH ranged from 5.08 to 9.10 and EC ranged from 26.25 to 2034 μs cm^−1^. A large range was also observed in soil texture fractions reflecting the diversity of the sampled area. The sand content ranged from 0.49% to 93.91%, silt ranged from 4.7% to 87.86%, and clay ranged from 1.38% to 31.73%. The variability of the soil properties can be attributed to the spatial variability of the sample set.

### 3.2. Optimization of Spectral Preprocessing and Modelling 

#### 3.2.1. Preprocessing Performance Evaluation

The prediction of SOM using all combination of preprocessing and modeling algorithms performed for the selection of best preprocessing algorithms for subsequent prediction of other soil properties. The R^2^_adj_ ranges 0.14 to 0.97 and 0.13 to 0.89 for calibration and validation dataset respectively (Table 6). The PLSR yielded R^2^_adj_ ranges 0.72 to 0.81 and 0.14 to 0.75 for calibration and validation, respectively (Table 6). The calibration and validation R^2^_adj_ for cubist ranges from 0.61 to 0.91 and 0.37 to 0.89, respectively. The R^2^_adj_ resulted using RF ranges 0.96 to 0.97 and 0.66 to 0.87 for calibration and validation, respectively, while calibration and validation R^2^_adj_ for ELM ranges from 0.14 to 0.75 and 0.13 to 0.81, respectively. The cubist appeared to be the best modelling algorithm with the highest validation R^2^_adj_ of 0.89; however, PLSR and RF also produced relatively high R^2^_adj_ of 0.84 and 0.87, respectively. The results showed the lower R^2^_adj_ for the validation than that of calibration dataset except few instances where 1st Derivative, 1st Derivative + Gap, 2nd Derivative + Gap, Savitzky-Golay + Gap and Savitzky-Golay used for preprocessing along with PLSR or ELM modeling algorithms. 

The 1st Derivative + Gap and 2nd Derivative + Gap were selected as the best performing preprocessing algorithms for further analysis, based on good R^2^_adj_ (>0.5) values, along with a general increase of R^2^_adj_ from calibration to validation (Figure 5). Although no improvement was seen from calibration to validation for SNV, it was also selected for further analysis based on consistently high R^2^_adj_ calibration values across all modelling algorithms.

In addition to the R^2^_adj,_ several other parameters were also calculated to finalize the decision. A list of performance indices is presented in Table 7 for SOM. Consistent performance of the models developed on the calibration dataset, internal validation dataset and external validation dataset was observed for the cubist model, while the highest performance was recorded in the results of RF model. ELM produced consistent low performance as observed in the values of each performance indicator (Table 7).

#### 3.2.2. Spectral Prediction for All Soil Properties

Based on R^2^_adj_ the SOM, silt and sand content were the best predicted soil properties, while VCS was the poorest (Table 8). The SOM, sand, silt, pH, and CS were all predicted very well using VIS-NIR spectroscopy with R^2^_adj_ greater than 0.60. The ms content predicted fairly with R^2^_adj_ of 0.53. However, the fs, clay, EC, and VCS were poorly predicted with R^2^_adj_ of 0.49, 0.26, 0.22, and 0.18 respectively. 

The highest prediction accuracy for the SOM was obtained using 1st Derivative + Gap as preprocessing and cubist as modeling algorithm. The RF produced highest prediction accuracy for the soil pH, sand, silt, ms, with 1st Derivative + Gap and second highest for SOM with 2nd Derivative + Gap as preprocessing algorithm. The highest prediction accuracy for soil CS content was obtained with PLSR while the prediction accuracy for ms content was as good as obtained by RF. The 1st Derivative + Gap was the most successful preprocessing algorithm yielding the best accuracy for all the soil properties except EC. The results depict 1st Derivative and RF as best preprocessing and modeling algorithms for this study.

## 4. Discussion

Soil properties were predicted with varying amounts of accuracy using VIS-NIR spectroscopy. Some of the soil properties could have been predicted better with inclusion of short-wave infrared (SWIR), such as ASD Field Spec series sensors (350 to 2500 nm). However, the spectroradiometer we used for this study has a spectral range of 342 to 2220 and the removal of edge effects lead to a further narrower spectral range (397 to 2212) for prediction model development. Though the advantages of this spectroradiometer include cost and ability to scan depth samples in-situ. When examining model prediction results it is important to note the occurrence of negative R^2^ values. Because 1 − [Sum of Squares Error (SSE)/Sum of Squares Treatment (SST)] was used to calculate R^2^_adj_, negative values are possible when model performance is very poor. In agreement with Islam et al. [15] SOM/SOC was the best predicted soil property with R^2^ value of 0.89. Research by Terra et al. [10] found SOC to correspond better in the MID infrared region of spectral data; however, they reported a lower R^2^ value (0.77) compared to the current research. The dark colour associated with SOM can be easily detected by broad absorptions in the visible region [6], which may explain why better predictions were seen in the current research compared to Terra et al. [10]. The current research found RF with 1st derivative + Gap to yield the best results. It is likely that 1st derivative + Gap performed the best due to its ability to enhance small spectral absorptions and increase predictive accuracy for complex data sets [52]. The machine learning approach that RF applies was also likely to help improve predictions. Islam et al. [15] and Terra et al. [10] both used PCA for modelling during their research. The differences achieved from different modelling algorithms is important to note for supporting the testing of several models in this study. 

Soil pH was strongly predicted in the current research with an external validation R^2^_adj_ of 0.63 when using RF for modelling. However, Reeves and McCarty [53] and Reeves et al. [54] both achieved higher R^2^ of 0.74 and 0.73, respectively, for the prediction of pH when using PLSR. In contrast, Terra et al. [10] achieved a lower R^2^ of 0.54 for the prediction of pH when using PCA. The EC was the one of most poorly predicted soil property in this research with an external validation R^2^_adj_ value of 0.22. Similarly, Islam et al. [15] also predicted EC with R^2^ of 0.10. Poor predictions of EC likely occurred for several reasons: (1) EC is strongly associated with water content and dry samples were used in this research; (2) Laboratory measured EC values are extremely low and would be difficult to pick up with the spectrometer; (3) VIS-NIR spectroscopy does not have enough energy to measure electronic transitions and it is likely that the unique spectral fingerprint of EC is present in the another area of the light spectrum [55]. 

Overall, the sand, silt and CS were well predicted with R^2^_adj_ of 0.70, 0.70 and 0.68, respectively, while other texture fractions were more poorly predicted (Table 5). In this study, RF was found to better predict sand, silt, and ms, when compared to PLSR, cubist and ELM algorithms. However, PLSR better predicted CS, while cubist better predicted fs content. A study by Hobley and Prater [56] also reported promising results for the prediction of texture fractions using VIS-NIR spectroscopy; however, contradictory to the current research they found PLSR to perform better than RF. Hobley and Prater [56], used Log10 transformation to invert their spectral data which could have an effect on the performance of modelling algorithms. A much smaller data set was also used in this study compared to the current research which also may affect the accuracy of model prediction. Conforti et al. [9] predicted sand and clay content with higher accuracy R^2^ of 0.81, and 0.83 respectively, while the silt content was predicted with similar prediction accuracy R^2^ of 0.70. Like Hobley and Prater [56], Conforti et al. [9] transformed the spectral data from reflectance to absorbance which may have influenced model accuracy. 

Clay content is generally well predicted by VIS-NIR spectroscopy with its unique absorptions fingerprint displaying around 1395, 1415, 2160 and 2208 nm for kaolinite, 2206 and 2230 nm for smectite and, 2206, 2340 and 2450 nm for illite clay minerals [24]. Contradictory to the current research sand and silt are generally more poorly predicted than clay. The poor prediction of clay in the current research is likely attributed to the lower clay content in these samples than those reported in the literature. The absence of SWIR region in the spectroradiometer used in this study also contribute to poor prediction of clay, as clay mineral have unique absorbance signature in this spectral range. Sand content is generally better predicted in the mid-IR region of the light spectrum; however, predictions can be seen in the VIS-NIR spectrum due to iron oxide contents on the sand grains [24]. 

Overall, the current research yielded promising results for the use of VIS-NIR spectroscopy to predict soil properties. This research demonstrated the use of several preprocessing and modelling algorithms when analyzing spectral data. The 1st Derivative + Gap was found to be the optimal preprocessing algorithm. Its ability to enhance small spectral absorptions and known benefits for complex data sets explain why 1st Derivative + Gap outperformed other preprocessing algorithms [52]. The 1st Derivative + Gap performed best in combination with RF as a modeling algorithm. The RF is known to work well with large amount of data and is quick in training [57]. The quick training of RF in combination with the enhanced spectral absorption from 1st Derivative + Gap likely contributed to the increase in prediction accuracy of soil properties using VIS-NIR spectroscopy. 

## 5. Conclusions

In conclusion, soil properties were predicted with varying degrees of success. The study demonstrated that VIS-NIR spectroscopy can be used to predict soil properties on air-dried ground samples for heterogenous soils of Ontario. However, it is not advanced enough to completely replace traditional sampling techniques. The findings of this study demonstrated the need to use several preprocessing and modelling algorithms when predicting soil properties with VIS-NIR spectroscopy as different algorithms performed differently depending on the soil property it was predicting. However, in general RF and 1st Derivative + gap can be labeled at the best combination of preprocessing and modelling algorithms.

## Figures and Tables

**Figure 1 sensors-21-06745-f001:**
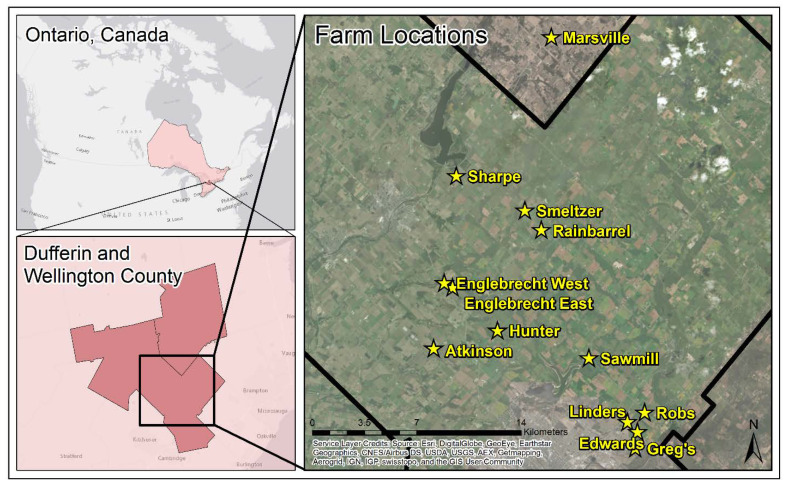
Location of the 13 farms selected for sampling within Dufferin and Wellington County, Ontario, Canada. The black line on the Farm Locations map represents the boundary of Wellington County. The yellow stars and yellow text represent the locations of the farms where soil samples were collected.

**Figure 2 sensors-21-06745-f002:**
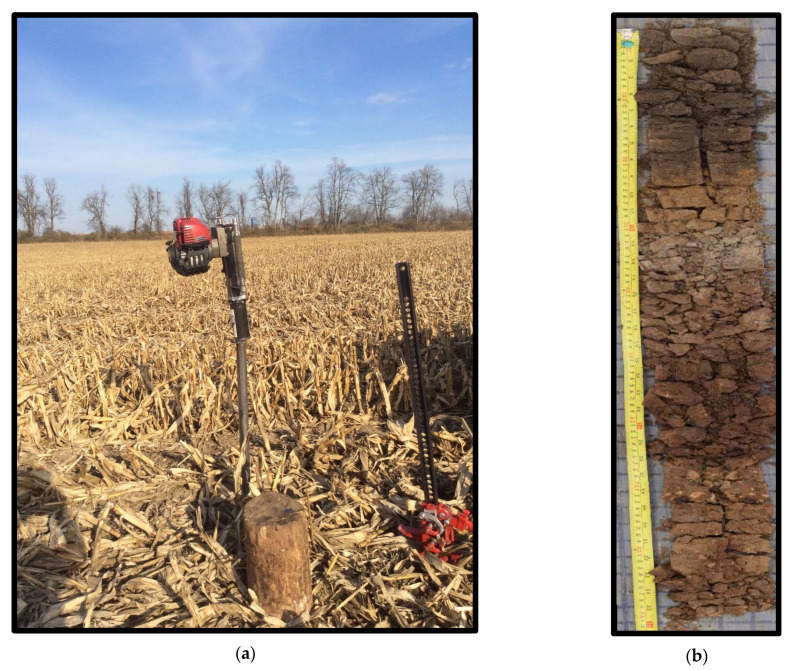
(**a**) Photo of modified post pounder (Deer Fence Canada, Dunrobin, ON, Canada) (**b**) a split core soil profile.

**Figure 3 sensors-21-06745-f003:**
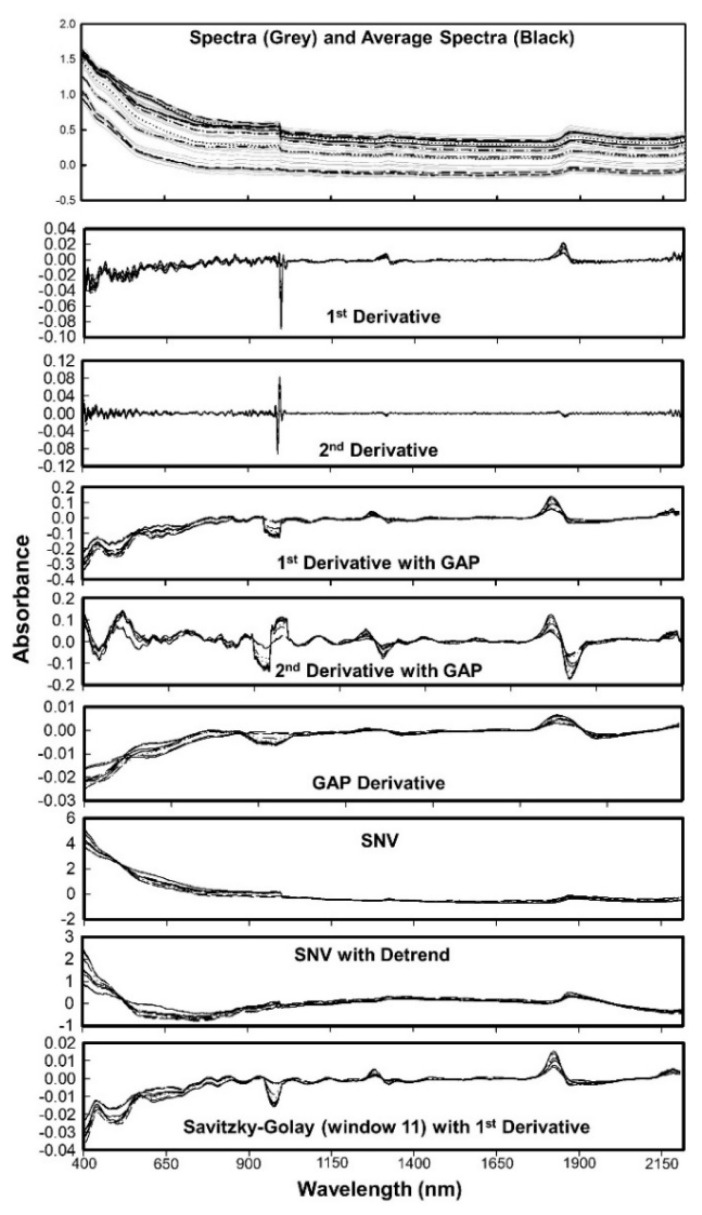
Spectra and the average spectra of first 10 soil samples used in this study. Effects of different preprocessing algorithms are also shown individually.

**Figure 4 sensors-21-06745-f004:**
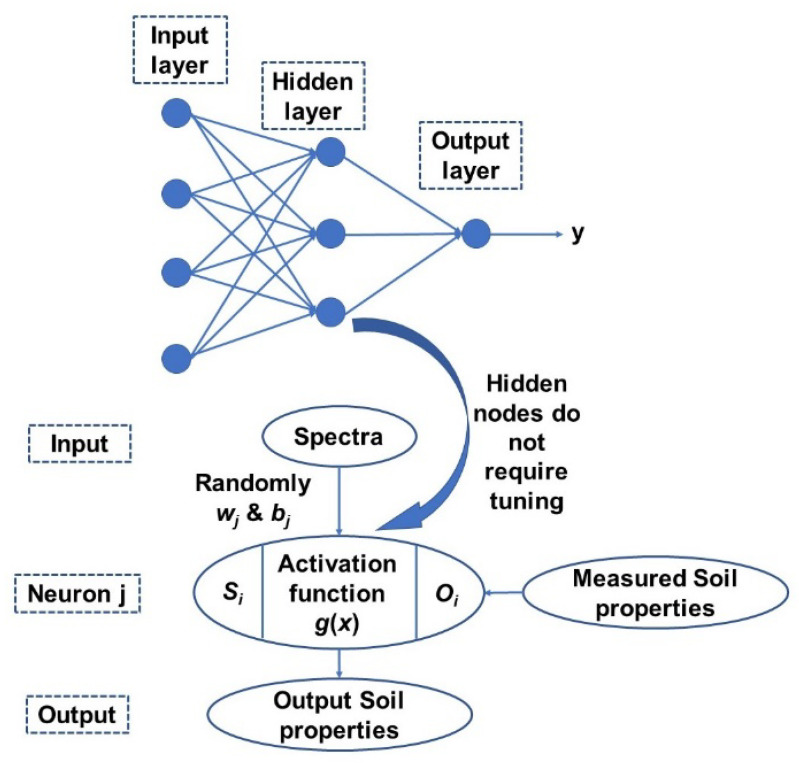
A general structure of the ELM model adopted in this study [46].

**Figure 5 sensors-21-06745-f005:**
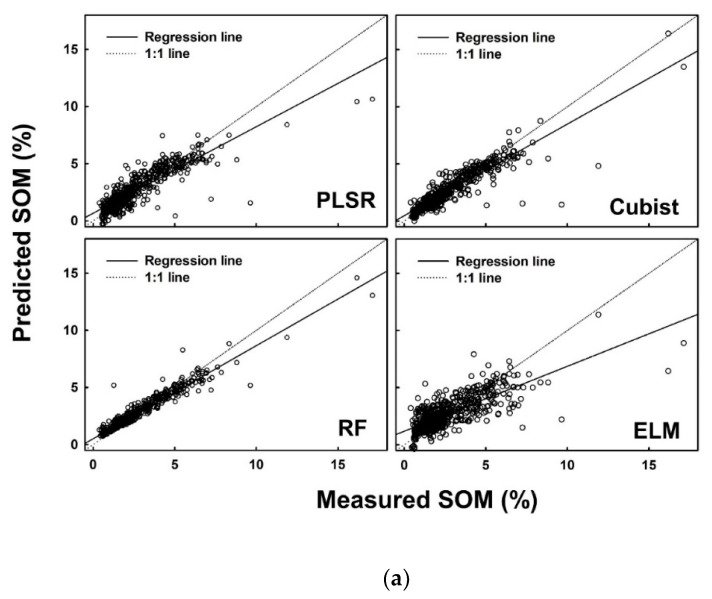
Measured versus predicted soil organic matter (%) using four different models (PLSR, Cubist, RF and ELM) for the (**a**) calibration dataset, (**b**) internal validation dataset, and (**c**) external validation dataset.

**Table 1 sensors-21-06745-t001:** Literature review of most commonly predicted soil properties using VIS-NIR spectroscopy with corresponding coefficient of determination (R^2^) on validation dataset.

References	Region	*n*	Model	*R*^2^*V*alidation					
				SOC/SOM	pH	EC	Sand	Clay	Silt
Johnson, et al. [7]	SSA	2845	PLSR	-	0.59	0.37	0.54	0.70	0.47
Gupta, et al. [8]	India	954	PLSR_LW_	0.70	-	-	0.72	0.61	-
Zhang et al. [2]	Canada	257	Cubist	0.66	0.67	0.12	0.50	0.70	0.00
Conforti, et al. [9]	Italy	267	PLSR	0.88	0.70	-	0.81	0.80	0.70
Terra, et al. [10]	Brazil	1259	SVM	0.65	0.24	-	0.89	0.86	-
Gholizade, et al. [11]	Malaysia	118	SMLR	0.81	0.59	0.51	-	-	-
P Leone, et al. [12]	Italy	374	PLSR	0.91	-	-	0.58	0.83	0.51
Lee, et al. [13]	USA	165	SMLR	-	-	-	0.76	0.80	0.80
Viscarra Rossel, et al. [14]	Australia	116	PLSR	0.72	0.73	0.29	0.75	0.67	0.52
Islam, et al. [15]	Australia	161	PCR	0.76	0.71	0.10	0.53	0.72	0.05

SSA: sub-Saharan Africa.

**Table 2 sensors-21-06745-t002:** Description of six preprocessing algorithms used in this study.

Preprocessing Algorithm	Impact	Equation
1st Derivative	Reduce the drift of the baseline and highlight some parts of the spectral information [38].	$FD\left(R\right)=\frac{R_{n+1}-R_{n}}{\lambda_{n+1}-\lambda_{n}}$
2nd Derivative	Reduce the drift of the baseline and liner trend. Also highlight some parts of the spectral information [38].	$SD\left(R\right)=\frac{FD_{n+1}-FD_{n}}{0.5\left(\lambda_{n+2}-\lambda_{n}\right)}$
Gap Derivative	Remove both additive and multiplicative effects. These methods enhance spectral resolution and eliminate background effects.	
Savitzky-Golay	Remove the high frequency noise from samples	
Standard Normal Variate (SNV)	It performs both the centeringand scaling together by subtracting the mean and normalizing with the standard deviation for each reflectance spectrum [38].	$SNV\left(R\right)=\frac{R-{µ}_{R}}{\sigma_{R}}$
Detrend	It involves fitting a 2nd order polynomial to the SNV transformed spectrum and subtracted from it to correct for wavelength dependent scattering effects	

**Table 3 sensors-21-06745-t003:** List of 13 preprocessing algorithms that were tested in combination with the 4 modelling algorithms.

Preprocessing	1st Derivative, 2nd Derivative, Gap Derivative, Savitzky-Golay, SNV, 1st Derivative + Gap, 2nd Derivative + Gap, Savitzky-Golay + Gap, Savitzky-Golay + 1st Derivative, Savitzky-Golay + 2nd Derivative, Savitzky-Golay + SNV, Savitzky-Golay + SNV + Detrend, SNV + Detrend
Modeling	Partial Least Square Regression (PLSR), Random Forest (RF), Cubist, Extreme Learning Machine (ELM)

**Table 4 sensors-21-06745-t004:** Brief description of the model performance indicators used in this study with their formula.

Indicator	Meaning	Formula
R^2^	Correlation coefficient of determination explains how well the variance of the spectral predicted values align with the lab measured values	$1-\frac{SS_{residuals}}{SS_{total}}$; $SS_{residuals}$ is the sum of squared of residuals or predicted, $SS_{total}$ is the total sum of squared
R^2^_adj_	Adjusted R^2^ or modified version of R^2^ adjusts for the number variables in the prediction model. While more predictor variables tend to increase (called overfitting) and often return an unwarranted high R^2^, adjusted R^2^ can determine how reliable the correlation is and how it is determined by the addition of more predictor variables. It compensates for addition of variables and only increase if the new variable enhances the model above what that would be obtained by chance.	$1-\frac{SS_{residuals}/\left(n-k\right)}{SS_{total}/\left(n-1\right)}$; $SS_{residuals}$ is the sum of squared of residuals or predicted, y-measured, x; $SS_{total}$ is the total sum of squared, *n* is the number of data points and k is the number of variables in the model.
CCC	Concordance correlation coefficient measuring the agreement between the measured and predicted values of soil properties or reproducibility or how close the predicted values are to the measured values (closeness to 1:1 line).	$\frac{2rs_{x}s_{y}}{{\left(\bar{x}-\bar{y}\right)}^{2}+s_{x}^{2}+s_{y}^{2}}$; r is the correlation coefficient, $\bar{x}$ is the mean of the measured, $\bar{y}$ is the mean of the predicted, $s_{x}^{2}$ variance of measured and $s_{y}^{2}$ is the variance of the predicted values.
MSE	Mean squared error measures the average squares of the error or the difference between predicted and measured values.	$\frac{1}{n}\overset{n}{\underset{i=1}{\sum }}{\left(y_{i}-x_{i}\right)}^{2}$; *n* is the number of data points, $y_{i}$ are the predicted values and $x_{i}\;$ are the measured values.
RMSE	Root mean squared error measures the difference between values predicted by a model and is the square root of the MSE.	$\sqrt{MSE}$
MSEc	Mean squared error of calibration dataset measuring how well the calibration worked	Same as MSE but for calibration dataset
RMSEc	Root mean squared error of calibration measuring how well the calibration worked	Same as RMSE but for calibration dataset
RPD	Ratio of performance of deviation or the ratio between the standard deviation of a variable and the standard error of prediction	$\frac{SD}{SEP}$; SD is the standard deviation of the sample $\sqrt{\frac{1}{n-1}}\overset{n}{\underset{i=1}{\sum }}{\left(y_{i}-\bar{y}\right)}^{2}$ and SEP is the standard error of prediction (calculated as RMSE)
RPIQ	Ratio of performance of interquartile distance is the interquartile range of the measured values divided by the RMSE	$\frac{IQ}{SEP}$; IQ is the interquartile range and SEP is the standard error of prediction (calculated as RMSE)

**Table 5 sensors-21-06745-t005:** Descriptive statistics for laboratory measured soil properties.

Properties	Mean	Median	Min	Max	σ	*n*
EC, μs cm^−1^	309.30	265.90	26.25	2034.00	197.86	1038
SOM, %	2.69	2.11	0.39	17.13	1.82	1025
pH	7.71	7.71	5.08	9.10	0.55	1041
Sand, %	45.11	41.85	0.49	93.91	20.20	238
Silt, %	43.23	45.46	4.70	87.86	17.07	238
Clay, %	11.67	10.68	1.38	31.73	6.23	238
VCS, %	3.69	2.06	0.03	41.29	5.57	208
CS, %	5.66	3.80	0.00	46.15	6.45	208
ms, %	15.57	12.28	0.91	69.94	10.96	208
fs, %	22.82	21.66	1.32	68.47	11.23	208

**σ** is standard deviation and, ***n*** is number of samples.

**Table 6 sensors-21-06745-t006:** The calibration and validation R^2^_adj_ resulting from all possible preprocessing and modeling algorithms for prediction of SOM.

Preprocessing Algorithms	Calibration R^2^_adj_				Validation R^2^_adj_			
	PLSR	Cubist	RF	ELM	PLSR	Cubist	RF	ELM
1st Derivative	0.81	0.84	0.97	0.45	0.75	0.79	0.79	0.62
1st Derivative + Gap	0.77	0.91	0.97	0.63	0.83	0.89	0.87	0.77
2nd Derivative	0.73	0.76	0.97	0.14	0.70	0.70	0.70	0.13
2nd Derivative + Gap	0.76	0.88	0.97	0.49	0.84	0.88	0.87	0.81
Savitzky-Golay + Gap	0.74	0.75	0.97	0.67	0.83	0.69	0.84	0.76
Gap Derivative	0.77	0.80	0.97	0.75	0.71	0.77	0.77	0.70
Savitzky-Golay	0.77	0.89	0.97	0.71	0.78	0.82	0.80	0.70
Savitzky-Golay + 1st Derivative	0.79	0.70	0.97	0.62	0.74	0.61	0.78	0.40
Savitzky-Golay + 2nd Derivative	0.78	0.61	0.97	0.40	0.68	0.37	0.71	0.29
Savitzky-Golay + SNV	0.72	0.92	0.96	0.28	0.64	0.76	0.75	0.20
Savitzky-Golay + SNV + Detrend	0.74	0.89	0.96	0.58	0.52	0.64	0.66	0.32
SNV	0.77	0.90	0.96	0.71	0.59	0.70	0.66	0.56
SNV + Detrend	0.78	0.90	0.96	0.57	0.59	0.65	0.71	0.26

**Table 7 sensors-21-06745-t007:** Model performance indicators of SOM prediction calculated in finalizing the right combination of preprocessing and modelling algorithms for all other soil properties.

		R^2^	CCC	MSE	RMSE	Bias	MSEc	RMSEc	RPD	RPIQ
PLSR	Calibration	0.76	0.86	0.88	0.94	0.00	0.88	0.94	2.03	2.46
	Validation	0.73	0.85	0.76	0.87	0.04	0.76	0.87	1.86	2.11
	External Validation	0.75	0.86	0.72	0.85	0.00	0.72	0.85	2.00	2.52
Cubist	Calibration	0.83	0.90	0.63	0.79	−0.08	0.62	0.79	2.41	2.91
	Validation	0.82	0.89	0.48	0.69	0.03	0.48	0.69	2.35	2.65
	External Validation	0.81	0.89	0.56	0.75	−0.08	0.55	0.74	2.27	2.87
RF	Calibration	0.94	0.95	0.29	0.54	0.01	0.29	0.54	3.53	4.28
	Validation	0.65	0.76	0.95	0.97	0.09	0.94	0.97	1.67	1.89
	External Validation	0.67	0.78	0.96	0.98	0.03	0.96	0.98	1.73	2.18
ELM	Calibration	0.57	0.72	1.57	1.25	0.00	1.57	1.25	1.52	1.84
	Validation	0.60	0.75	1.13	1.06	0.23	1.08	1.04	1.53	1.73
	External Validation	0.60	0.76	1.17	1.08	0.11	1.16	1.08	1.57	1.98

**Table 8 sensors-21-06745-t008:** The validation R^2^_adj_ for various soil properties using selected best preprocessing and four modeling algorithms.

Properties.	1st Derivative + Gap				2nd Derivative + Gap				SNV			
	A	B	C	D	A	B	C	D	A	B	C	D
SOM, %	0.83	0.89	0.87	0.77	0.84	0.88	0.87	0.81	0.59	0.70	0.66	0.56
EC, μs cm^−1^	−0.02	0.00	−0.02	−0.02	−0.01	−0.03	−0.03	0.22	−0.01	−0.03	−0.03	0.22
pH	0.57	0.62	0.63	0.52	0.48	0.54	0.53	0.48	0.48	0.54	0.53	0.48
Sand, %	0.48	0.47	0.70	0.53	0.29	0.40	0.46	0.45	0.29	0.40	0.46	0.45
Silt, %	0.46	0.53	0.70	0.60	0.40	0.39	0.42	0.25	0.4	0.39	0.42	0.25
Clay, %	0.13	0.26	0.20	0.19	0.23	0.20	0.25	0.25	0.23	0.20	0.25	0.25
VCS, %	0.18	−0.02	0.17	0.04	0.11	0.00	0.02	−0.01	0.11	0.00	0.02	−0.01
CS, %	0.68	0.08	0.15	0.46	0.30	0.58	0.22	0.02	0.30	0.58	0.22	0.02
ms, %	0.50	0.24	0.53	0.39	0.31	0.28	0.32	0.09	0.31	0.28	0.32	0.09
fs, %	−0.01	0.49	−0.02	−0.02	0.01	0.03	0.14	−0.01	0.01	0.03	0.14	−0.01

A: PLSR; B: Cubist; C: RF; and D: ELM modeling algorithms.

## Data Availability

The spectroscopic and soil laboratory data used in this manuscript is not publicly available and may be available upon request with a signed data sharing agreement.

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
