# Peer review of "Evaluation of Optimized Preprocessing and Modeling Algorithms for Prediction of Soil Properties Using VIS-NIR Spectroscopy"

_sensors, 2021, doi:10.3390/s21206745_

Round 1
Reviewer 1 Report
This manuscript describes work that tries to evaluate the performance of different preprocessing algorithms (single and in combination) when doing empirical modeling on spectral + chemical data for soil samples. While the overall objective is sound, and perhaps the actual analysis is too, the description of the methodology is significantly lacking, to the point that the results are not interpretable.
Focusing on section 2.4, this is where you would give a concise description of the technical methodology that was used. But, not a single equation is presented, not one algorithm sequence is shown, and not a single spectral plot is provided. It is all text.
You should give technical description of the methodology that was used, and provide specific and concise elaboration about all the modeling and preprocessing algorithms that you used in this work. Assume the reader does not know about PLSR, and give them an explanation so they could understand what you did. Same for the other modeling algorithms. The "one-liners" descriptions you gave for each modeling algorithm are insufficient at the least.
Moreover, what is "adjusted R-squared"? Give the equation you used as this is a crucial component of the study. Why would you use it instead of a regular R-squared? You write "considers the number of samples", why is that important? Why does it matter? What was the consideration to use this?
There are more issues with the paper, and you can fine my comments in the PDF (attached).
Hence, I will recommend major revisions, and hope you will address the raised points.

Author Response
Reviewer 1: Response to the comments
This manuscript describes work that tries to evaluate the performance of different preprocessing algorithms (single and in combination) when doing empirical modeling on spectral + chemical data for soil samples. While the overall objective is sound, and perhaps the actual analysis is too, the description of the methodology is significantly lacking, to the point that the results are not interpretable.
Response: Thank you so much for your valuable comments. We have added some details on the preprocessing and modeling algorithms with a figure on the modeling structure of ELM. We have cited the detailed methodology as well. We have also added more results and interpreted them carefully to learn from the study.
Focusing on section 2.4, this is where you would give a concise description of the technical methodology that was used. But, not a single equation is presented, not one algorithm sequence is shown, and not a single spectral plot is provided. It is all text.
You should give technical description of the methodology that was used and provide specific and concise elaboration about all the modeling and preprocessing algorithms that you used in this work. Assume the reader does not know about PLSR and give them an explanation so they could understand what you did. Same for the other modeling algorithms. The "one-liners" descriptions you gave for each modeling algorithm are insufficient at the least.
Response: Thank you so much. In order to maintain the brevity of the manuscript, we did not include more details on the methodology. However, we have expanded the methodology with equations with descriptions. In fact, we have also included a figure showing the general structure of the ELM methodology. We have included a table with preprocessing algorithms and their equations. We have also included a figure showing some examples of the collected raw spectra, the impact of various preprocessing algorithms. Again, thanks for your comment. It helped us to greatly improve this section.
Moreover, what is "adjusted R-squared"? Give the equation you used as this is a crucial component of the study. Why would you use it instead of a regular R-squared? You write "considers the number of samples", why is that important? Why does it matter? What was the consideration to use this?
Response: Thank you very much for your comments. First, we have included several other model performance indicators in the revised version including adjusted R-square. We have included details on each of these indicators in a new table along with their calculation/equation. We have provided justification for using each of these indicators including adjusted r-squared.
There are more issues with the paper, and you can fine my comments in the PDF (attached).
Response: Thank you so much for your detailed comments on the manuscript. We have carefully considered edit/comment by comment and revised the manuscript. There was so much, and we did not include them here. However, we have provided a track-changed version of the manuscript showing the changes made in the manuscript.
Hence, I will recommend major revisions, and hope you will address the raised points.
Reviewer 2 Report
This research tests different combinations of preprocessing and modelling techniques for the estimation of soil properties based on VIS-NIR spectroscopy measurements. An extensive soil sampling campaign provides an excellent basis for the experimental setup. Methods from common R packages are exploited which increases the relevance of the results. Overall, the manuscript is well written and the results seem relevant, however, several parts need extensive revision and elaboration: 1) The state of the art is missing relevant studies, 2) materials (sensor used) and methods (preprocessing techniques and parameters) lack important information and details, 3) results are only provided in most rudimentary form. They lack quantitative error metrics and could benefit from spectral plots, as well as summary figures such as bar plots of the accuracy metrics. 4) the discussion misses to address a sensor related perspective and the general assessment of transferability. More specific point are adressed below:
Abstract
Line 20: Notation of “vis-NIR” or “VIS-NIR” should be unified throughout the manuscript. The explicit spectral data ranges considered in the analysis should be expressed here as well. Especially, as most commonly NIR spectroscopy includes the spectral range until 2400 or 2500 nm.
Introduction
Line 47/49: Please make sure to have consistent spacing before adding the reference and after a period. E.g., at Line 47 “surface[5]” and Line 49 “properties[5].The”
Line 71: I’m missing the absorbance transformation and continuum removal (e.g., see Dotto et al. 2018, cited below) for the listing and discussion of commonly used preprocessing techniques in soil spectroscopy/pedometrics. Also, Vasques et al. (2008) is missing as a relevant reference for the evaluation of spectral preprocessing techniques.
Vasques, G.M., Grunwald, S., Sickman, J.O., 2008. Comparison of multivariate methods for inferential modeling of soil carbon using visible/near-infrared spectra. Geoderma 146, 14–25. https://doi.org/10.1016/j.geoderma.2008.04.007.
Line 98: “We could not find a study which used combinations of preprocessing and modelling algorithms on a single spectral data set for the prediction of soil properties.” Although not directly the same, several studies previously combined the study of different preprocessing and modelling techniques for the estimation of soil properties. e.g., the following references that are missing from the state of the art and the discussion of the results and should be included in the manuscript:
Dotto, A.C., Dalmolin, R.S.D., ten Caten, A., Grunwald, S., 2018. A systematic study on the application of scatter-corrective and spectral-derivative preprocessing for multivariate prediction of soil organic carbon by Vis-NIR spectra. Geoderma 314, 262–274. https://doi.org/10.1016/j.geoderma.2017.11.006
Gholizadeh, A., Borůvka, L., Saberioon, M.M., Kozák, J., Vašát, R., Němeček, K., 2015. Comparing different data preprocessing methods for monitoring soil heavy metals based on soil spectral features. Soil and Water Research 10 (2015), 218–227. https://doi.org/10.17221/113/2015-SWR
Nawar, S., Buddenbaum, H., Hill, J., Kozak, J., Mouazen, A.M., 2016. Estimating the soil clay content and organic matter by means of different calibration methods of vis-NIR diffuse reflectance spectroscopy. Soil and Tillage Research 155, 510–522. https://doi.org/10.1016/j.still.2015.07.021
Methods
Line 167: Formula of OM (%) is missing a “0” at the end for %?
Line 178: Please include a more detailed description of the sensor/spectrometer used in this study (“Ocean Optics USB400 spectrometers”). E.g., on spectral range, spectral resolution / FWHM, size/geometry of pixel array, lens, signal-to-noise ratio. Also, the addition of some exemplary soil spectra is recommended, as it would allow a visual assessment of the spectral properties.
Line 186: A short summary on the preprocessing methods is missing in the methods part (e.g., in chapter 2.4 “Optimization of Data Processing”). Also, the parameters selected for preprocessing methods should be included in the methods part (e.g., window size for Savitzky-Golay and gap-segment smoothing), as these can strongly impact modeling results.
Line 192: The stated number of 13 preprocessing methods is misleading. More precise: 13 combinations of 6 methods included in the prospectr package (1st, 2nd derivative, Savitzky-Golay, gap-segment smoothing, SNV and Detrend) are tested. Also in this part, the effect of the preprocessing methods should be visualized in spectral plots of the same exemplary soil sample.
Line 219: In addition to the R², common quantitative error metrics used in soil spectroscopy/pedometrics such as the RMSE, and the RPIQ (Ration of Performance to Interquartile Distance) of the validation dataset should be added and discussed to allow an improved and more quantitative assessment of the results.
Results
Table 4 and 5 As stated above RMSE and RPIQ of the validation data should be added to the table and discussed further on. Calibration R² is not important and could be removed to make some space. In addition to the table (bar) plot figures could improve the readability of the results. E.g., Modelling technique vs. R²/RMSE and preprocessing technique vs. R²/RMSE
Line 254: Table 4: The “1st Derivative” entry is double in the table (first and last line).
Discussion
A critical discussion regarding the effect on the spectral sensor used in this study (“Ocean Optics USB400”) is missing in this section. Especially regarding the spectral resolution and the spectral region covered by the instrument. Major differences might be introduced due to the missing clay absorptions. Also, the general transferability to more commonly used sensors that should be discussed.
Line 319: “Clay content is generally well predicted by vis-NIR spectroscopy with its unique absorptions fingerprint displaying around 7000 nm [12].” Major clay absorptions are found in the SWIR 2 between 2.1-2.4 µm (depending on clay mineral, e.g., see cited reference [12] Rossel & Behrens (2010))
Author Response
Reviewer 2: Response to the comments
This research tests different combinations of preprocessing and modelling techniques for the estimation of soil properties based on VIS-NIR spectroscopy measurements. An extensive soil sampling campaign provides an excellent basis for the experimental setup. Methods from common R packages are exploited which increases the relevance of the results. Overall, the manuscript is well written and the results seem relevant, however, several parts need extensive revision and elaboration: 1) The state of the art is missing relevant studies, 2) materials (sensor used) and methods (preprocessing techniques and parameters) lack important information and details, 3) results are only provided in most rudimentary form. They lack quantitative error metrics and could benefit from spectral plots, as well as summary figures such as bar plots of the accuracy metrics. 4) the discussion misses to address a sensor related perspective and the general assessment of transferability.
Response: Thank you very much for your detailed review and constructive comments. These have been helpful in improving the quality of the paper. In general, we have updated the paper with some recent literature, drastically improved the materials and methods including sensor and the data analysis methods, modified results with new information and description, modified discussion to improve the quality of the paper. We have included a track-changed version of the manuscript as well to identify the changes made. For more details, we have responded to your comments point-by-point below.
More specific point are addressed below:
Abstract
Line 20: Notation of “vis-NIR” or “VIS-NIR” should be unified throughout the manuscript. The explicit spectral data ranges considered in the analysis should be expressed here as well. Especially, as most commonly NIR spectroscopy includes the spectral range until 2400 or 2500 nm.
Response: Thanks a lot for the comment. We have revised the manuscript following your comment.
Introduction
Line 47/49: Please make sure to have consistent spacing before adding the reference and after a period. E.g., at Line 47 “surface[5]” and Line 49 “properties[5].The”
Response: Thanks. We have corrected these in the revised version.
Line 71: I’m missing the absorbance transformation and continuum removal (e.g., see Dotto et al. 2018, cited below) for the listing and discussion of commonly used preprocessing techniques in soil spectroscopy/pedometrics. Also, Vasques et al. (2008) is missing as a relevant reference for the evaluation of spectral preprocessing techniques.
Vasques, G.M., Grunwald, S., Sickman, J.O., 2008. Comparison of multivariate methods for inferential modeling of soil carbon using visible/near-infrared spectra. Geoderma 146, 14–25. https://doi.org/10.1016/j.geoderma.2008.04.007.
Response: Thanks a lot. We have added additional preprocessing methods and corresponding references.
Line 98: “We could not find a study which used combinations of preprocessing and modelling algorithms on a single spectral data set for the prediction of soil properties.” Although not directly the same, several studies previously combined the study of different preprocessing and modelling techniques for the estimation of soil properties. e.g., the following references that are missing from the state of the art and the discussion of the results and should be included in the manuscript:
Dotto, A.C., Dalmolin, R.S.D., ten Caten, A., Grunwald, S., 2018. A systematic study on the application of scatter-corrective and spectral-derivative preprocessing for multivariate prediction of soil organic carbon by Vis-NIR spectra. Geoderma 314, 262–274. https://doi.org/10.1016/j.geoderma.2017.11.006
Gholizadeh, A., Borůvka, L., Saberioon, M.M., Kozák, J., Vašát, R., Němeček, K., 2015. Comparing different data preprocessing methods for monitoring soil heavy metals based on soil spectral features. Soil and Water Research 10 (2015), 218–227. https://doi.org/10.17221/113/2015-SWR
Nawar, S., Buddenbaum, H., Hill, J., Kozak, J., Mouazen, A.M., 2016. Estimating the soil clay content and organic matter by means of different calibration methods of vis-NIR diffuse reflectance spectroscopy. Soil and Tillage Research 155, 510–522. https://doi.org/10.1016/j.still.2015.07.021
Response: Thank you so much for your comment. We have referred the relevant studies and cite them as well in the manuscript. Again, thanks for your suggestions.
Methods
Line 167: Formula of OM (%) is missing a “0” at the end for %?
Response: Thanks. We have corrected this in the revised version.
Line 178: Please include a more detailed description of the sensor/spectrometer used in this study (“Ocean Optics USB400 spectrometers”). E.g., on spectral range, spectral resolution / FWHM, size/geometry of pixel array, lens, signal-to-noise ratio. Also, the addition of some exemplary soil spectra is recommended, as it would allow a visual assessment of the spectral properties.
Response: Thank you very much. We have added additional information about the spectrometer and sensors in the revised version.
Line 186: A short summary on the preprocessing methods is missing in the methods part (e.g., in chapter 2.4 “Optimization of Data Processing”). Also, the parameters selected for preprocessing methods should be included in the methods part (e.g., window size for Savitzky-Golay and gap-segment smoothing), as these can strongly impact modeling results.
Response: Responded together with the next comment.
Line 192: The stated number of 13 preprocessing methods is misleading. More precise: 13 combinations of 6 methods included in the prospectr package (1st, 2nd derivative, Savitzky-Golay, gap-segment smoothing, SNV and Detrend) are tested. Also in this part, the effect of the preprocessing methods should be visualized in spectral plots of the same exemplary soil sample.
Response: In order to maintain the brevity of the manuscript, we did not include more details on the methodology. However, we have expanded the methodology with equations with description. In fact, we have also included a figure showing the general structure of the ELM methodology. We have included a table with preprocessing algorithms and their equations. We have also included a figure showing some examples of the collected raw spectra, the impact of various preprocessing algorithms. Again, thanks for your comment. It helped us to greatly improve this section.
Line 219: In addition to the R², common quantitative error metrics used in soil spectroscopy/pedometrics such as the RMSE, and the RPIQ (Ration of Performance to Interquartile Distance) of the validation dataset should be added and discussed to allow an improved and more quantitative assessment of the results.
Response: Thank you so much for your comments. In fact, we have calculated a series of model performance indicators. We have added a table listing all the indicators and their values for SOM prediction. We have also used internal validation and external validation in this study. We have documented these in the table and presented the result with a figure showing the performance of 4 different models for each of the three datasets: calibration, internal validation, and external validation.
Results
Table 4 and 5 As stated above RMSE and RPIQ of the validation data should be added to the table and discussed further on. Calibration R² is not important and could be removed to make some space. In addition to the table (bar) plot figures could improve the readability of the results. E.g., Modelling technique vs. R²/RMSE and preprocessing technique vs. R²/RMSE
Response: Thank you very much for the comment. We agree that having RMSE and RPIQ could be added to these tables. However, we have added a new table of several performance indicators including RMSE and RPIQ. We have listed them separately in a new table then adding to these tables. However, we only used all the criteria at the optimization phase and did not use them for other soil properties. We have also included measured vs predicted SOM graphs for each model and three datasets; calibration, internal validation, and external validation. Hope this additional information brings added value to the paper.
Line 254: Table 4: The “1st Derivative” entry is double in the table (first and last line).
Response: Thanks. We have deleted this from the table.
Discussion
A critical discussion regarding the effect on the spectral sensor used in this study (“Ocean Optics USB400”) is missing in this section. Especially regarding the spectral resolution and the spectral region covered by the instrument. Major differences might be introduced due to the missing clay absorptions. Also, the general transferability to more commonly used sensors that should be discussed.
Response: Thanks for the comment. We have added a quick discussion on the spectral instrument. However, as we did not use multiple instruments, it is not possible to direct the comparison of results. However, the potential or perceived impact of the spectrometer is indicated briefly in the revised manuscript.
Line 319: “Clay content is generally well predicted by vis-NIR spectroscopy with its unique absorptions fingerprint displaying around 7000 nm [12].” Major clay absorptions are found in the SWIR 2 between 2.1-2.4 µm (depending on clay mineral, e.g., see cited reference [12] Rossel & Behrens (2010))
Response: Thanks for the comment. We have corrected and updated this in the revised manuscript and added the new references.
Round 2
Reviewer 2 Report
The revised manuscript has significantly improved compared to the previous version. The methods are overall much better described and missing results metrics and figures were added. However, some minor proofreading is still necessary, some inconsistencies removed and important sensor related aspects need to be added to the discussion.
Please check again the missing whitespaces. There are still plenty missing, mostly before references, e.g., just on page 2 Line 61, 64, 69, 74, 76…
Line 20 The first sentence of the abstract still needs to be rephrased. It is no grammatically correct sentence (missing verb). And also, a strange sentence to begin the abstract with. Also, the spectral range is missing the last “0”.
Line 200, Fig 3 The new Figure 3 shows absorbance spectra, but in the manuscript only reflectance is mentioned. If the reflectances have been transformed to absorbance before applying the prediction modelling, this step needs to be added to the methods part. Or otherwise, if reflectances were used, then they should be shown in this figure.
Line 208 According to the manufacturer guide (https://www.hamamatsu.com/resources/pdf/ssd/mini-spectro_kacc0002e.pdf, e.g., figure4), the C9914GB has a spectral resolution of 6 nm, maybe it is only binned to 4nm? Please check.
Line 308 The sentence here “The spectral data was split into a 70% calibration set and a 30% validation set” indicates a split in two parts, but later on in Table 7 “Calibration, Validation and External Validation” are presented. It needs to be clarified how the samples were split and used for validation.
Line 411-416 “3) VIS-NIR spectroscopy does not have enough energy to measure 414 electronic transitions and it is likely that the unique spectral fingerprint of EC is present 415 in the another area of the light spectrum[24]. “ The soil property Electric Conductivity (EC) is mainly an indicator of soil solute (cation or anion) related to salinity. It has been successfully (R² > 0.8) modelled in other studies using laboratory VNIR-SWIR spectroscopy of more saline soils (e.g., Farifteh et al. 2007, doi:10.1016/j.rse.2007.02.005). So, its prediction is definitely possible from VNIR-SWIR spectra, but certainly depends on soil mineralogy. Furthermore, reference [24]: Viscarra Rossel & Behrens (2010) do not relate to EC.
Line 431 It is good that the part on clay absorption was added to the manuscript. However, what is still missing, is a critical discussion on the spectrometers used in this study and how their properties (e.g., spectral coverage, resolution, SNR?) might affect the results compared to sensors that cover the full SWIR range until 2500. Certainly, the relatively low clay prediction is a consequence of the lack in 2.2. µm coverage. Also, pH might be predicted more accurate from the 2.3 µm carbonate abortion. The relatively low spectral resolution compared to the most used ASD fields spectrometer is also missing from the discussion. On the other hand, the sensors used in this study are much less expensive, and still give useful results. These aspects need to be added especially for a submission to a journal named sensors.
Author Response
Reviewer 2: Response to comments
The revised manuscript has significantly improved compared to the previous version. The methods are overall much better described and missing results metrics and figures were added. However, some minor proofreading is still necessary, some inconsistencies removed and important sensor related aspects need to be added to the discussion.
Response: Thanks for the comment, we have improvised the manuscript and fixed the minor revisions suggested by the reviewer.
Please check again the missing whitespaces. There are still plenty missing, mostly before references, e.g., just on page 2 Line 61, 64, 69, 74, 76…
Response: Thanks for the comment. We have fixed the space issue at all the in-text citations.
Line 20: The first sentence of the abstract still needs to be rephrased. It is no grammatically correct sentence (missing verb). And also, a strange sentence to begin the abstract with. Also, the spectral range is missing the last “0”.
Response: Thanks for the comment. We have modified the sentence to “The absorbance spectra for air-dried and ground soil samples from Ontario, Canada were collected in the visible and near-infrared (VIS-NIR) region from 343 to 2200 nm.”
Line 200: Fig 3 The new Figure 3 shows absorbance spectra, but in the manuscript only reflectance is mentioned. If the reflectances have been transformed to absorbance before applying the prediction modelling, this step needs to be added to the methods part. Or otherwise, if reflectances were used, then they should be shown in this figure.
Response: Thanks for the comment. We used the soil absorbance spectra collected in the VIS-NIR region 343 to 2200 nm. We have added the following sentences to section 2.4 for clarification. “The spectra signatures (absorbance) associated with each wavelength then used for further processing. We used the spectral data in absorbance format since it reduces nonlinearity and shows higher correlation with soil properties [2,14,38,39].”
We did not transform the absorbance spectra to reflectance. Ji et al. 2016 and Vasava et al.2019 mentioned that better relationship between soil constituent and reflectance can be achieved by transforming reflectance to absorbance. Soil reflectance can be transformed to absorbance units using either Lambert-Beer’s law
Or by Kubelka-Munk equation
Line 208: According to the manufacturer guide (https://www.hamamatsu.com/resources/pdf/ssd/mini-spectro_kacc0002e.pdf, e.g., figure4), the C9914GB has a spectral resolution of 6 nm, maybe it is only binned to 4nm? Please check.
Response: Thank you very much for the comment. We checked that spectrometer model, data generated by the spectrometer (raw files), and the dataset used in our study, and yes, the spectral resolutions are 6 nm. However, at the NIR range, we have resampled the spectra at 4 nm to process further. We have modified this in the manuscript.
Line 308: The sentence here “The spectral data was split into a 70% calibration set and a 30% validation set” indicates a split in two parts, but later on in Table 7 “Calibration, Validation and External Validation” are presented. It needs to be clarified how the samples were split and used for validation.
Response: Thanks for the comment, we have added an explanation “The initial spectral data was split into a 70% calibration set and a 30% validation set by Kennard and Stone method [52], and the 30% dataset was kept separate as external validation dataset. The calibration spectral dataset was further divided into a 70% calibration and 30% as cross-validation or internal validation dataset. The calibrated model was separately tested for external validation dataset.”
Line 411-416: “3) VIS-NIR spectroscopy does not have enough energy to measure electronic transitions and it is likely that the unique spectral fingerprint of EC is present in the another area of the light spectrum [24]. “The soil property Electric Conductivity (EC) is mainly an indicator of soil solute (cation or anion) related to salinity. It has been successfully (R² > 0.8) modelled in other studies using laboratory VNIR-SWIR spectroscopy of more saline soils (e.g., Farifteh et al. 2007, doi:10.1016/j.rse.2007.02.005). So, its prediction is definitely possible from VNIR-SWIR spectra, but certainly depends on soil mineralogy. Furthermore, reference [24]: Viscarra Rossel & Behrens (2010) do not relate to EC.
Response: Thanks for the comment. Yes, we agree that electrical conductivity (EC) is mainly an indicator of soil solute. The literature review we did [Gholizade et al. 2013 (R2 = 0.51), Zhang et a. 2017 (R2 = 0.12), Islam et al. 2003 (R2 = 0.10), Shibusawa et al. 2001 (R2 = 0.65), Viscarra Rossel et al. 2006 (R2 = 0.29), Soriano-Disla et al. 2017 (R2 = 0.25)] made us conclude that prediction of EC is challenging. We have corrected the references as well.
Line 431: It is good that the part on clay absorption was added to the manuscript. However, what is still missing, is a critical discussion on the spectrometers used in this study and how their properties (e.g., spectral coverage, resolution, SNR?) might affect the results compared to sensors that cover the full SWIR range until 2500. Certainly, the relatively low clay prediction is a consequence of the lack in 2.2. µm coverage. Also, pH might be predicted more accurate from the 2.3 µm carbonate abortion. The relatively low spectral resolution compared to the most used ASD fields spectrometer is also missing from the discussion. On the other hand, the sensors used in this study are much less expensive, and still give useful results. These aspects need to be added especially for a submission to a journal named sensors.
Response: Thanks. Yes, we agree that the inclusion of SWIR could have improved the prediction of some of the soil properties. However, the spectroradiometer we used for this study has a spectral range of 342 to 2220 and the edge noise leads to spectral range further narrow.
We have added the following sentences to the discussion “Some of the soil properties could have been predicted better with the inclusion of short-wave infrared (SWIR), such as ASD Field Spec series sensors (350 to 2500 nm). However, the spectroradiometer we used for this study has a spectral range of 342 to 2220 and the removal of edge effect leads to the spectral range for prediction model development further narrower. Though the advantages of this spectroradiometer include cost and ability to scan depth samples in-situ.”